# A Determination Method for Gait Event Based on Acceleration Sensors

**DOI:** 10.3390/s19245499

**Published:** 2019-12-12

**Authors:** Chang Mei, Farong Gao, Ying Li

**Affiliations:** Artificial Intelligence Institute, Hangzhou Dianzi University, Hangzhou 310018, China; changmei103@163.com (C.M.); liyinglxy@163.com (Y.L.)

**Keywords:** inertial sensors, gait recognition, event determination, acceleration signal, threshold segmentation, frequency domain integration

## Abstract

A gait event is a crucial step towards the effective assessment and rehabilitation of motor dysfunctions. However, for the data acquisition of a three-dimensional motion capture (3D Mo-Cap) system, the high cost of setups, such as the high standard laboratory environment, limits widespread clinical application. Inertial sensors are increasingly being used to recognize and classify physical activities in a variety of applications. Inertial sensors are now sufficiently small in size and light in weight to be part of a body sensor network for the collection of human gait data. The acceleration signal has found important applications in human gait recognition. In this paper, using the experimental data from the heel and toe, first the wavelet method was used to remove noise from the acceleration signal, then, based on the threshold of comprehensive change rate of the acceleration signal, the signal was primarily segmented. Subsequently, the vertical acceleration signals, from heel and toe, were integrated twice, to compute their respective vertical displacement. Four gait events were determined in the segmented signal, based on the characteristics of the vertical displacement of heel and toe. The results indicated that the gait events were consistent with the synchronous record of the motion capture system. The method has achieved gait event subdivision, while it has also ensured the accuracy of the defined gait events. The work acts as a valuable reference, to further study gait recognition.

## 1. Introduction

Gait disorders are usually associated with the ageing population as well as falling, leading to both a reduced quality of life and an increased mortality rate. The detection of gait events is an important tool in clinics [1,2], including human activity recognition for healthcare [3,4] and motor recovery assessments for effective rehabilitation strategies [5,6]. Hence, it is essential to develop an effective algorithm for the accurate detection of gait events.

Gait locomotion, as a fundamental activity for all humans, is a cyclic spatiotemporal complex act. Gait information can be acquired by collecting kinematics signals, bioelectrical signals, videos and images [7,8,9,10,11,12,13]. In traditional gait analysis methods, a three-dimensional motion capture (3D Mo-Cap) system and force plate pressure signal can accurately describe the 3D motion of the human body, while can accurately detect the gait event with error of 0.13% [14], which is often called the golden standard method in gait analysis [15,16,17,18]. However, the cost of Mo-Cap system equipment and the force measuring board have a high cost, and there are limitations of areas such as a laboratory environment for data acquisition, while it is also unable to synchronously detect human daily activities. In addition, the 3D Mo-Cap system is also affected by light and other environmental factors [17,19], limiting the widespread use of this method. The time of a heel strike (HS) and toe off (TO) can be determined by the plantar switching signal and, consequently, the gait can be divided into support phase and swing phase [20]. However, the disadvantage of the foot switch signal is that it can only output discrete 0, 1 signals, unable to further subdivide the gait pattern. The mechanical signals, based on force plate, are also widely used to analyze gait locomotion [21,22,23]. Empirical thresholds can be used to determine the time of HS and TO. However, due to the limitation of the number of force-measuring plates, it is impossible to measure the gait of several consecutive cycles, while the empirical threshold in this method does not have the characteristics of adaptability.

Considering the development of wireless networks and wearable sensing technology, lightweight, small-size, lower power consumption, portable and low-cost wearable sensors are now widely used and indispensable in gait analysis [24,25,26]. Researchers employed wearable sensors to accurately estimate the support and swing periods of healthy adults and amputees, while using the force plate signal to verify the feasibility of the results. It is noted that wearable sensors can effectively distinguish the gait pattern [27]. In recent years, wearable sensors have been widely used in gait speed estimation [28] and the assessment of a user’s health state, while being based on gait abnormalities [29] and fall detection [30]. Some studies recognize and evaluate the daily activities through Microsoft Kinect sensors which measured concurrently with a 3D Mo-Cap system (gold standard), with the result showing that the Microsoft Kinect sensors can effectively identify the characteristics of patients’ daily activities and, hence, that it can be used as an effective clinical evaluation method [31,32]. Wearable devices are also widely used to track human walking activities [33]. However, wearable technologies have inherent limitations, and their sensor responses are often influenced by wearers’ behavior, motion, clothing and environmental factors [34]. In a word, it is assumed that gait recognition, relying on inertial data acquired by commonly used smart devices, has become reasonable and has been recently addressed by many research groups.

Acceleration signals, collected by inertial sensors, have been widely used to divide the two gait events of HS and TO [35,36,37]. Studies have proved that the threshold of comprehensive change rate of the acceleration signal can be used to determine HS and TO events, but the final result will be affected by the degree of smoothing filtering of acceleration signals [38]. Some studies utilized a spatial filtering method to determine HS events and TO events, based on acceleration signals. They proved that the value of cut-off frequency for signal filtering is relatively difficult to determine, while different cut-off frequencies will influence the result [39]. Human gait phase has diversity and complexity. The two gait phases can be divided into several gait periods with different characteristics [40,41,42,43]. In recent years, some researchers have used the threshold of comprehensive acceleration to preliminarily segment the signal. Then, based on the principle of local maximum of acceleration signal, after filtering, the gait events are divided into four basic events [44], i.e., HS, toe strike (TS), heel off (HO), and TO. Nevertheless, the selection of threshold will affect the initial segmentation effect, while the cut-off frequency of filtering will affect the determination of local maximum.

Although the use of the triaxial accelerometers to determine human gait events is becoming increasingly mature, there are relatively few detailed studies on the four events of gait. In this study, a gait event segmentation method is presented, based on acceleration signals. In the case of continuous walking, the respective acceleration signal is automatically extracted, while the above four continuous events are effectively determined.

## 2. Acquisition and Processing of Acceleration Signal Data

### 2.1. Data Acquisition

Five healthy, able-bodied subjects (five males: age, 24–26 years; height, 170–180 cm; weight 60–75 kg) participated in the experiments. Considering the characteristics of foot movement and ground-touching/ground-leaving events, during walking, two triaxial acceleration modules are used in signal acquisition. Each subject wore sensors on their vertical position of right heel and the horizontal position of the toe. The acceleration signals, in the three directions, were collected by two triaxial accelerometers with a sampling frequency of 1000 HZ, while two marker points were placed at the heel and toe. Using Vicon’s 3D Mo-Cap, and analysis system (Oxford Metrics Limited, Oxford, UK), real-time acquisition of 3D position signals was achieved, at a sampling frequency of 100 HZ. The acceleration signal and 3D Mo-Cap signal were transmitted to the computer, running the acquisition software, through real-time communication between the synchronous console and the computer. In order to eliminate the influence of gait speed on gait event detection, the experiment arranged for the subjects to walk on the Track Master TMX428CP treadmill and maintain a uniform speed of 1.2 m/s. In the acquisition process, Vicon synchronously acquired 3D Mo-Cap signals of heel and toe, which could then be used to verify and analyze the results of gait event determination (contrast results).

In this study, the sensors (Data LINK ACL300, Biometrics Ltd., Newport, UK) have 32 synchronous digital acquisition channels, with a sampling frequency of 1000 Hz. It is a sensor with motion artifact suppression (patented) that can be freely moved, while the sensor directly transmits data wirelessly (Figure 1).

### 2.2. Data Processing

In the process of experimental signal acquisition, the acceleration is easily influenced by the noise of surrounding environment, of the acquisition equipment itself and by physiological aspects, causing the acquired raw acceleration signal to be distorted, to a certain degree. In this paper, the wavelet modulus maximum is used to eliminate noise in the raw acceleration signal. First, the acceleration signal needs to be processed by discrete wavelet [45]. Second, when the wavelet scale changes, the modulus maxima of the real signal are proportional to the wavelet scale, while the modulus maxima of the noise are inversely proportional to the wavelet scale, as the real signal and noise are separated. The details are as follows:

For a function f(x), n is a negative integer, n≤β≤n+1. If exist polynomials of n order $p_{n}(h)$ and constant A while $h_{0}>0$, for any $h\leq h_{0}$, the following relation is satisfied:(1)$$\left|f(x_{0}+h)-p_{n}(h)\right|\leq A{\left|h\right|}^{\beta }$$
where, β is the singular point in the signals. Let Wa(s,i) be the discrete wavelet transform of noise signal a(i), at position i on scale s. For any point i in a certain field $\left(i_{0}-\delta ,i_{0}+\delta \right)$ of $i_{0}$, the following is satisfied:(2)$$\left|Wa(s,i)|\;\leq \;|Wa(s_{0},i_{0})\right|$$

If $\frac{\partial Wa(s_{0},i)}{\partial x}$ has a zero crossing at $i=i_{0}$, the point $\left(s_{0},i_{0}\right)$ is called local extremum. The $\left(s_{0},i_{0}\right)$ is the modulus maximum point of the wavelet transform. The relationship between wavelet scale j and signal index β is as follows:(3)$$\log_{2}|Wa(2^{j},i)|\leq \log_{2}k+j\beta$$
where, $k\in R^{n}$, in Equation (3), when β>0, the maximum value of wavelet modulus increases with the increase of scale. When β<0, the maximum value of wavelet modulus decreases with the increase of scale. Generally, the noise index β<0, the signal index β>0. Finally, after removing the modulus maxima of noise, the acceleration signal is reconstructed with the remaining modulus maxima, while the noise-free (filtered) acceleration signal is obtained. The comparison between heel vertical acceleration signal $a_{Hz}$ and toe vertical acceleration signal $a_{Tz}$, before and after noise elimination is illustrated in Figure 2 and Figure 3. It is evident that the effect of noise elimination is ideal.

## 3. Methods of Gait Event Analysis

### 3.1. Gait Event

The center of gravity of the human body swings back and forth in the front, back, left and right directions, during human walking. Alternating movement of feet, throughout the gait cycle, is as shown in Figure 4, where R represents the right foot. In a complete gait cycle, the right foot experiences four basic gait events: HS, TS, HO, TO. Figure 4 shows that the gait can be divided into supporting phase and swinging phase, during the time of HS and TO. Some studies point out that the acceleration of the heel and toe reaches the maximum, on the time of HS and TO [38]. Therefore, the occurrence of HS and TO events is often determined by finding out the moment where the heel and toe acceleration is maximum. At the same time, when the heel position is minimized as the HS event occurs, the toe position gradually starts to rise as the TO event occurs. Therefore, in some research, the heel and toe position signals, collected by a 3D motion capture system, are often used to determine HS and TO events [25]. It is pointed out in the literature [46] that, HS and TS events occur at the lowest position of heel and toe, respectively. Also, when HO and TO events occur, heel and toe displacement increases sharply.

Regarding the determination of gait events, studies point out that HS and TO events occur near the local maximum of the heel vertical acceleration signal, the events of HS and TO are identified if the differential of heel vertical acceleration signal is equal to 0 [38]. The final result will be affected, to some extent, by the smoothing filtering of the acceleration signal. The HS event occurred between the data after the local maximum and the maximum values of the vertical acceleration signal [25]. Some studies preliminarily segment the signal, based on the threshold of the acceleration signal, while then detecting the HS and TS events, using the local maximum of the Butterworth high-pass filtered heel vertical acceleration signal [44], which points out that HO and TO events occur after HS and TS events and before the local maximum value of the heel vertical acceleration signal. However, the initial segmentation effect will be affected by the threshold value, while the determination of the local maximum value will be affected by the filter cut-off frequency.

The advantages of the algorithm, presented in the literature [25,38,44], are combined, while a simple algorithm for determining the four events of HS, TS, HO and TO is proposed. The position evolution is obtained by quadratic integration of the acceleration signals of the heel and toe. Considering continuous walking, four continuous gait events are automatically extracted. The algorithm is mainly divided into two parts: (1) initial segmentation of continuous heel acceleration signal $a_{H}$ and toe acceleration signal $a_{T}$, based on the threshold of comprehensive change rate of the acceleration signal; (2) in the segmented acceleration signal, four basic gait events are determined by the displacement signal, as computed from the quadratic integration of the heel and toe acceleration signal.

After eliminating the noise in the acceleration signal, in order to remove the influence of the gravity component on the acceleration signal, the average value of the vertical acceleration signal on the whole continuous signal needs to be subtracted from the vertical acceleration signal (acceleration signal in the *z*-axis direction):(4)$$a_{\mathrm{Sz}}(i)=a_{\mathrm{Sz}}{\left(i\right)}_{orignal}-\sum_{i=1}^{N}a_{\mathrm{Sz}}(i)/N$$
where, $a_{\mathrm{Sz}}(i)$ is the vertical acceleration signal after removing the gravity component. $a_{\mathrm{Sz}}{\left(i\right)}_{orignal}$ is the raw acceleration signal in the vertical direction. For the heel acceleration signal, the subscript S corresponds to H; for the toe acceleration signal, the subscript S corresponds to T and N is the total number of continuous acceleration signals.

### 3.2. Preliminary Segmentation

When humans are walking, four stages are experienced: (i) heel strike; (ii) heel strike and toe strike; (iii) heel off and toe strike; (iv) heel off and toe off. At this time, the acceleration signal of heel and toe experience, from low to high and from high to low, while the variation of acceleration signal is increasingly intense, as the accelerometer is almost static, when the feet are all in contact with the ground. The threshold of the acceleration signal’s integrated rate is used to preliminarily segment the continuous acceleration signal, an approach which is compared to the segmentation method, based on the comprehensive acceleration threshold, as presented in the literature [15]. The results show that the proposed method is easier, for the initial segmentation of the continuous acceleration signal, while the effect is also very improved. It is not affected by the spikes in the signal, such as the peak of the acceleration signal.

The derivative of acceleration refers to the acceleration signal’s integrated rate, which represents the comprehensive change rate of acceleration in three directions, used to express the degree of acceleration change. This is calculated as follows:(5)$$Ja_{S}=\sqrt{{\left(\frac{dax_{{}_{S}}}{dt}\right)}^{2}+{\left(\frac{day_{{}_{S}}}{dt}\right)}^{2}+{\left(\frac{daz_{{}_{S}}}{dt}\right)}^{2}}$$
where, $Ja_{S}$ is the comprehensive rate of acceleration change, $ax_{{}_{S}},ay_{{}_{S}},az_{{}_{S}}$ are the acceleration values in the three directions. Similarly, the subscript S corresponds to H in the heel acceleration signal, the subscript S corresponds to T in the toe acceleration. Since the collected acceleration data is discrete, the derivative is solved in the form of difference, as follows:(6)$$Ja_{S}(i)=\sqrt{\begin{array}{l} \left(ax_{S}(i+1)-ax_{S}(i){)}^{2}+(ay_{S}(i+1\right)\\ -ay_{S}(i){)}^{2}+(az_{S}(i+1)-az_{S}(i){)}^{2} \end{array}}$$

In order to reduce the error of segmentation, in this paper, the size window W is used to smooth the comprehensive rate of acceleration change, in order to achieve the effect of smooth filtering, thus eliminating the small peak value of the comprehensive rate of acceleration change. The calculation is as follows:(7)$$ja_{S}(j)=\frac{1}{W}\sum_{i}^{i-W+1}Ja_{S}(i)$$
where, $ja_{S}(j)$ is the comprehensive change rate of acceleration, after smooth filtering, while the window size, considered in this study, is 30 ms [37].

After smoothing filtering of the comprehensive rate of acceleration change, the appropriate threshold is selected and the acceleration signal is processed in sections. The main purpose is to determine the moment when the accelerometer is close to static, fully supported on the ground. Using the following Equation (8) to make a preliminary division of gait, a binary function of heel and toe acceleration signals is produced, when the accelerometer is close to the stationary state, the appropriate phase is a flat phase, the binary function is 0, while the non-flat phase is the non-stationary state of the accelerometer and the binary function is equal to 1.
(8)$$Ta_{S}(i)=\left\{\begin{array}{l} 0\;ja_{S}(i)\leq th_{S}\;\\ 1\;ja_{S}(i)>th_{S} \end{array}\right.$$

In Equation (8), similarly, for the signal of the heel position, the subscript S corresponds to H; for the signal of the toe position, the subscript S corresponds to T; for the selection of threshold $th_{S}$, $th_{S}=r\times max(ja_{S}(i))$ is considered [47], while r is generally determined by the specific acceleration signals, as collected from different individuals. Actually, the choice of threshold does not have much effect on the final result, because Equation (8) is only for a preliminary gait segmentation, while the ultimate goal is to accurately identify the four basic events of gait. Next, based on the initial segmented acceleration signal, the following algorithm is used to continue to determine the time of HS, TS, HO and TO four events.

### 3.3. Gait Event Determination

The double integrals of the acceleration signal provides the displacement [48,49]. There are two main methods to convert the acceleration signal into displacement: time domain integration and frequency domain integration. The time domain integration is mainly the dynamic correspondence between time and acceleration signal, whereas frequency domain integration is basically the integration of the signal in the frequency domain, through the Fourier transform. Actually, the displacement, obtained by the quadratic integration of acceleration in the time domain, is far from the theoretical ideal. On the contrary, the frequency domain integration is relatively stable, but it also faces the existence of trend term, after the quadratic integration. In this study, the displacement signal of heel and toe is obtained by quadratic integration of the acceleration signal of heel and toe in frequency domain, while the wavelet decomposition coefficient threshold method is used to filter the displacement, removing the trend term, generated by the double integration.

It is known that the acquired acceleration signal is the Fourier component of $a_{\mathrm{sz}}(t)$, at any frequency w, given by:(9)$$a_{\mathrm{sz}}(t)=A_{sz}(k)e^{jw}{}^{t}$$

The displacement from the acceleration signal, through double integration is as follows:(10)$$d_{\mathrm{sz}}(t)=\int_{0}^{t}(\int_{0}^{t}A_{\mathrm{sz}}(k)e^{jw}{}^{t}dt)dt=\frac{A_{\mathrm{sz}}(k)}{(jw{)}^{2}}e^{jw}{}^{t}$$

The displacement after discretization is:(11)dsz(n)=1N∑k=0N−1Asz(k)(jw)2ejwk(2πnkN)
where, w=2πkf, f is the sampling frequency.

Due to the influence of gravity and jitter on the process of experimental acquisition, the average acceleration is generally not zero, while the displacement, after the quadratic integration, often includes a significant displacement trend term, which greatly affects the accuracy of the heel and toe displacement. At the same time, the acceleration signal, under the continuous gait period, is collected in the experiment and the final displacement produces a large offset. In this study, the wavelet decomposition coefficient threshold is used to remove the integrated trend term. The db6 wavelet basis was selected, to perform 8-layer wavelet decomposition of the integrated displacement signal while, according to the wavelet coefficient decomposition graph of displacement, the wavelet of trend term is removed by the wavelet soft threshold method.

#### 3.3.1. Determination of Time of Heel Strike (HS) Event

The heel acceleration signal is divided into flat phase and non-flat phase. The flat phase is the state in which the heel is in contact with the ground, while the non-flat phase is the period in which the heel swings in the air. In our study, the HS event is defined as, in the preliminary segmentation of the heel acceleration signal, the first trough where the heel displacement signals drop sharply for the first time.
(12)$$T_{\mathrm{HS}}=t_{H\min1}$$

In Equation (12), $t_{\mathrm{Hmin}1}$ is the moment when the heel displacement is at its first minimum value within the piecewise signal.

#### 3.3.2. Determination of Time of Toe Strike (TS) Event

Similarly, the TS event is defined as, in the preliminary segmentation of the toe acceleration signal, the first trough where the toe displacement signals drop sharply for the first time. Thus, the time of TS event is:(13)$$T_{\mathrm{TS}}=t_{\mathrm{Tmin}1}$$

In Equation (13), $t_{\mathrm{Tmin}1}$ is the moment when the toe displacement signals drop sharply to first trough.

#### 3.3.3. Determination of Time of Heel Off (HO) Event

HO event occurs after TS event. In our study, HO event is defined as, in the preliminary segmentation of the heel acceleration signal, the first trough where the heel displacement signals rise sharply for the first time.
(14)$$T_{\mathrm{HS}}=t_{H\min2}$$

In Equation (14), $t_{\mathrm{Hmin}2}$ is the moment when the heel displacement signals rise sharply to first trough.

#### 3.3.4. Determination of Time of Toe Off (TO) Event

The TO event is defined as, in the preliminary segmentation of the toe acceleration signal, the first trough where the toe displacement signals rise sharply for the first time.
(15)$$T_{\mathrm{TO}}=t_{T\min2}$$

In Equation (15), $t_{\mathrm{Tmin}2}$ is the moment when the toe displacement signals rise sharply to first trough.

## 4. Results and Discussion

### 4.1. Analysis of Preliminary Segmentation

Based on the threshold of the comprehensive rate of acceleration change $ja_{H}$ and $ja_{T}$, the vertical acceleration signal $a_{\mathrm{Hz}}$ of the heel and the vertical acceleration signal $a_{\mathrm{Tz}}$ of the toe are preliminarily segmented in order to generate the binary functions $Ta_{H}$ and $Ta_{T}$ of the acceleration signal of the heel and toe.

In Figure 5, the blue solid line is the acceleration singal, the black dotted line is the binary function, and the red solid line is the comprehensive change rate of acceleration. As shown in Figure 5, based on the threshold value of the comprehensive rate of acceleration signal change, a binary function of acceleration signal segments generated, according to which, the acceleration signal is divided into flat phase and non flat phase, determining four basic gait events, in the preliminary segmentation of heel signal and toe signal. Implementing Equations (12)–(15) in the above algorithm, four basic events (HS, TS, HO, TO) are established.

### 4.2. Analysis of Heel and Toe Displacement

According to Equations (9)–(11), the vertical acceleration of heel and toe is integrated twice, in the frequency domain, to calculate the position change of heel and toe (Figure 6 and Figure 7).

Figure 6 and Figure 7 show the position change of heel and toe without detrending, This is because of filtering of the signal to eliminate noise due to the instability of the accelerometer and the influence of gravity in the experiment. Meanwhile, the experiment collects several consecutive gait periods, therefore, the displacement, generated by the quadratic integration, will produce a significant displacement trend term, which affects the accuracy of the results severely. In this study, the detrending of the displacement is based on the wavelet decomposition threshold. Figure 8 shows 8-level wavelet decomposition of heel and toe displacement.

In this study, the trend term, obtained by frequency domain integration, is removed according to the wavelet decomposition soft threshold method. The 8-level wavelet decomposition of heel and toe displacement, using Sym8 wavelet base, produces results as illustrated in Figure 8. It is evident that the trend migration of heel and toe displacement is caused by ca8. A soft threshold method is used to obtain the filtered heel and toe displacement, by setting ca8 to 0, while the other layers remain the same. Figure 9 and Figure 10 show the vertical displacement of the heel and toe after filtering. It is shown that the wavelet decomposition filtering completely eliminates the trend migration of heel and toe displacement.

### 4.3. Analysis of Gait Event Determination

The vertical displacement of heel and toe is calculated above, using Equations (12)–(15), while the times of occurrence of HS, TS, HS and TO are determined respectively, as shown in Figure 11 and Figure 12.

In Figure 11 and Figure 12, the abscissa is the time, while the ordinate is the displacement signal of heel and toe, after quadratic integration. The gait continuously experienced four stages: (i) HS event, (ii) TS event and HS event, (iii) HO event and TS event, (iv) TO event, as entering the swing period. The presented algorithm demonstrates obvious features at each event point and thus realizes the determination of gait events. The HS event is the first time the heel contacts the ground, which occurs at the first trough of heel displacement, in the segmented signal, while the TS event is accompanied by the first contact of the toe with the ground, which occurs at the first trough of toe displacement, in the segmented signal. At the occurrence of the HO event, heel displacement begins to rise sharply, which occurs at the second trough of heel displacement. At the occurrence of the TO, with the toe off the ground, toe displacement rises sharply, which is something occurring at the second trough of toe displacement.

### 4.4. Error Analysis and Consistency Test

Finally, the 3D Mo-Cap signals of heel and toe, obtained synchronously, during the experiment, are compared and verified, while statistical error analysis and Bland–Altman consistency test are carried out.

Figure 13 illustrates the error statistical analysis, using the proposed algorithm and the results of 3D Mo-Cap signal (contrast results). The time difference between the four gait events, as determined in this study, and the time recorded by Vicon, is mainly about +20 ms, which is basically consistent with the normal distribution.

Figure 14 shows a Bland–Altman consistency test of four basic gait events, compared to the time recorded by 3D Mo-Cap system. The upper and lower dotted lines, in the figure, are the upper and lower limits of the 95% consistency limit, while the points beyond the limit represent the result points with larger error. The solid line in the middle is the mean value of the time difference recorded with the 3D Mo-Cap system. In these graphs, the abscissa is the mean value of the results of the theoretical analysis and the contrast results, while the ordinate is the difference between the contrast result and the result of the theoretical analysis. In the Bland–Altman test of the HS event, there are two points beyond the following limit, indicating that 6.67% points are within the 95% consistency limit, while in the Bland–Altman of the TS event, there is no point beyond the limit, indicating that the determination of the TS event is more accurate. The Bland–Altman test of the HO event and the TO event, produced only one point out of range, while the proportion of points out of range is as low as 3.33%. The difference between the results obtained in this paper and the control results is actually acceptable, while they are also in good accordance with the results recorded by Vicon.

## 5. Conclusions

In conclusion, based on the accuracy of detection of gait events, recognition rate and computational complexity, a series of methods of gait event detection for activity monitoring and fall detection is presented. Considering the rapid development of wearable electronic devices, inertial sensors, such as accelerometers, are fast becoming widely used in gait analysis, due to their portability and low-cost characteristics. The foot acceleration signal contains abundant gait feature information, reflecting the change of spatial position and speed, during human motion, while effectively describing the motion state of lower limbs. The respective device can be worn on the body and can transmit wirelessly through the gait signal acquisition and the transmission of a triaxial accelerometer module. Its simple and practical application characteristics show that acceleration signal is an ideal signal source in gait event detection. In this study, the threshold segmentation algorithm of the comprehensive rate of acceleration signal change is used. First, the initial segmentation of the acceleration signal is completed, then, the vertical acceleration of the heel and toe is integrated twice, to derive the displacement change. Due to the displacement, generated by the quadratic integration, a strong displacement trend term is produced, which greatly affects the accuracy of the results. To continue to detrend the displacement, the wavelet decomposition threshold is used. According to the characteristics of vertical displacement change of heel and toe, the four gait events are determined in the preliminary segmented signal. According to error analysis and consistency testing, the proposed determination of gait events provides results consistent with those recorded by the 3D Mo-Cap system. Compared with the traditional method of the gold standard, the wearable sensor is not only convenient for collecting data, but also can synchronously monitor human activities in daily life. In addition, the 3D Mo-Cap system is more sensitive to light and various environmental factors, while having strict requirements for the experimental site. The above work has certain theoretical and practical value to encourage further research on gait classification and recognition.

## Figures and Tables

**Figure 1 sensors-19-05499-f001:**
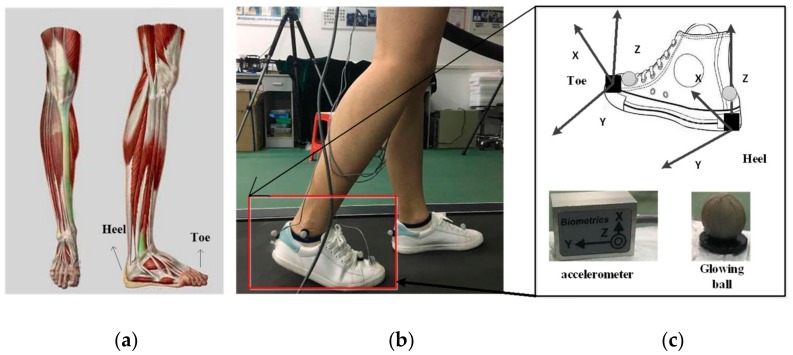
Hardware module and acquisition process diagram of experimental acquisition equipment. (**a**) The position of heel and toe in lower extremity; (**b**) data acquisition process; (**c**) equipment for acceleration signal.

**Figure 2 sensors-19-05499-f002:**
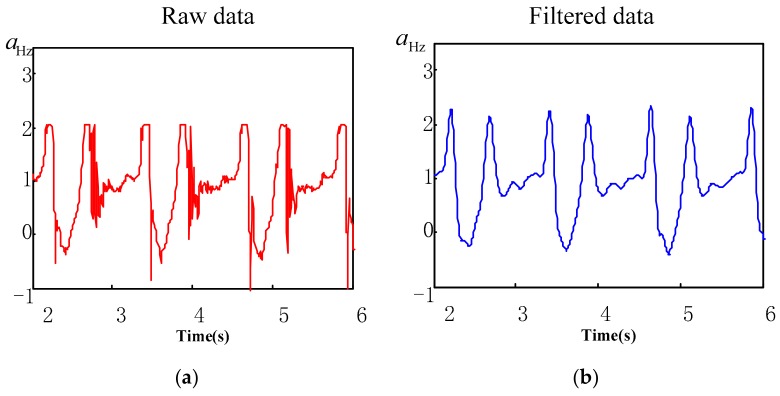
Contrast between the heel vertical acceleration signal states, before and after noise elimination. (**a**) Raw heel acceleration signal; (**b**) noise-free heel acceleration signal.

**Figure 3 sensors-19-05499-f003:**
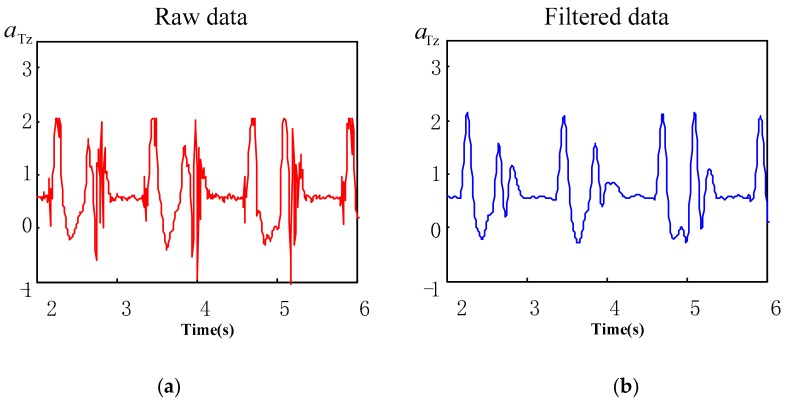
Contrast between the toe vertical acceleration signal states, before and after noise elimination. (**a**) Raw toe acceleration signal; (**b**) noise-free toe acceleration signal.

**Figure 4 sensors-19-05499-f004:**
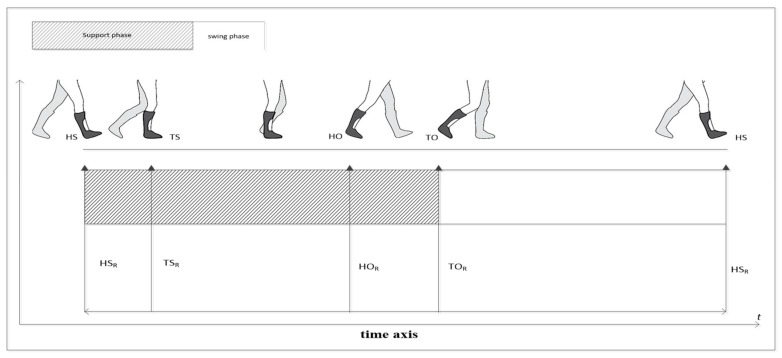
The schematic diagram of lower limb movement throughout the gait cycle.

**Figure 5 sensors-19-05499-f005:**
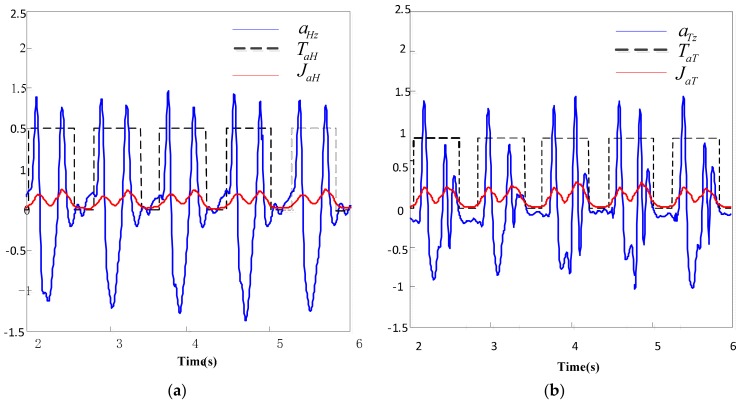
Preliminary segmentation results of the heel and toe signals. (**a**) Preliminary segmentation result of the heel signals; (**b**) preliminary segmentation results of the toe signals.

**Figure 6 sensors-19-05499-f006:**
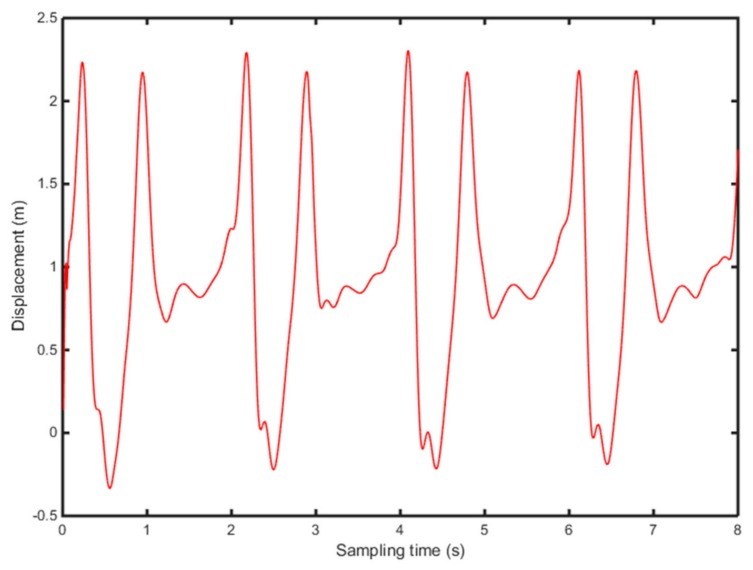
Displacement change of the heel, without filtering.

**Figure 7 sensors-19-05499-f007:**
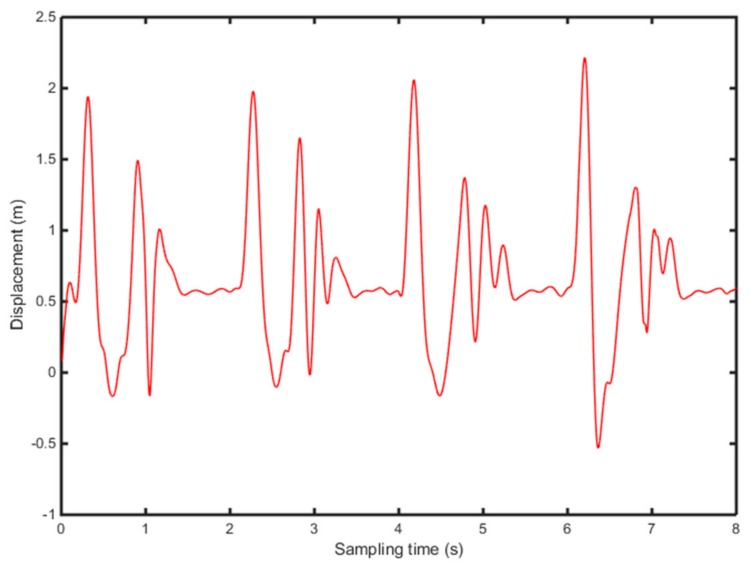
Displacement change of the toe, without filtering.

**Figure 8 sensors-19-05499-f008:**
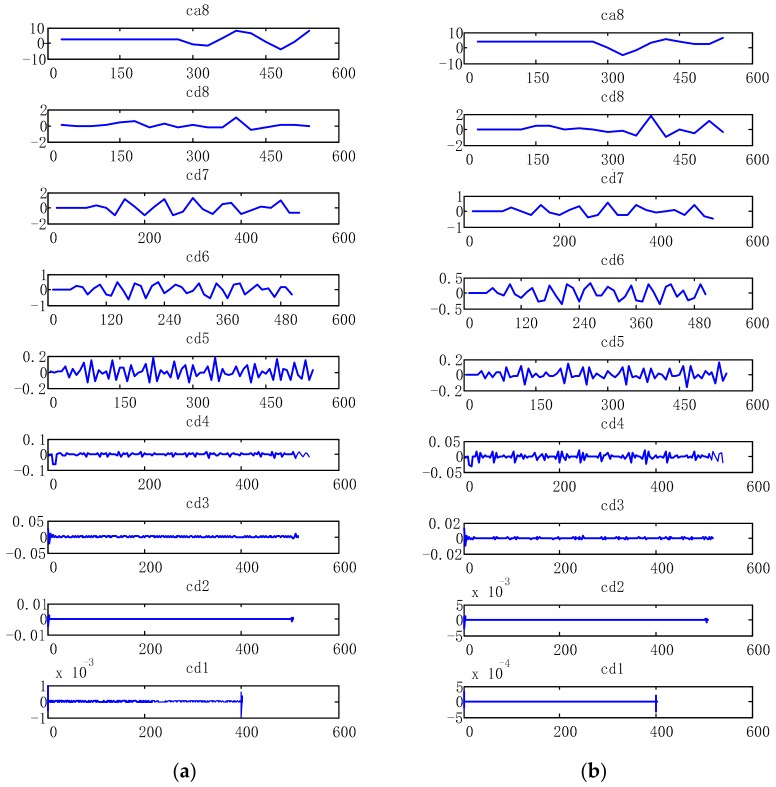
Wavelet decomposition of heel and toe displacement. (**a**) Wavelet decomposition of the heel displacement; (**b**) wavelet decomposition of the toe displacement.

**Figure 9 sensors-19-05499-f009:**
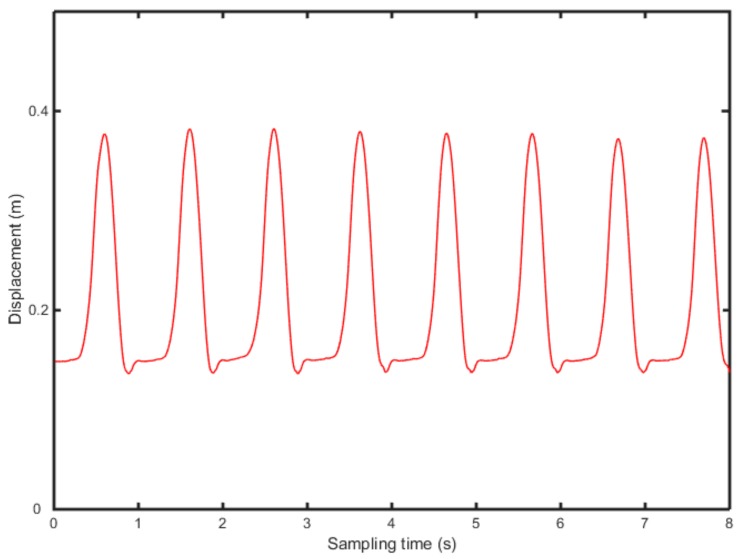
Displacement change of the heel with filtering.

**Figure 10 sensors-19-05499-f010:**
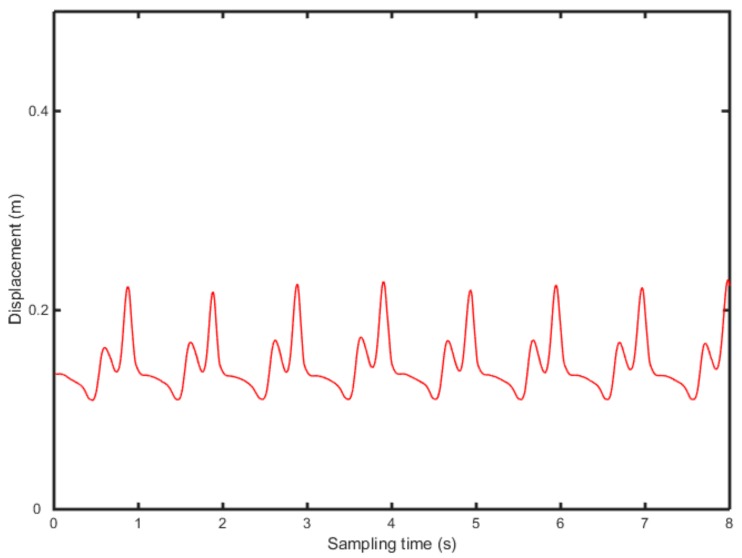
Displacement change of the toe without filtering.

**Figure 11 sensors-19-05499-f011:**
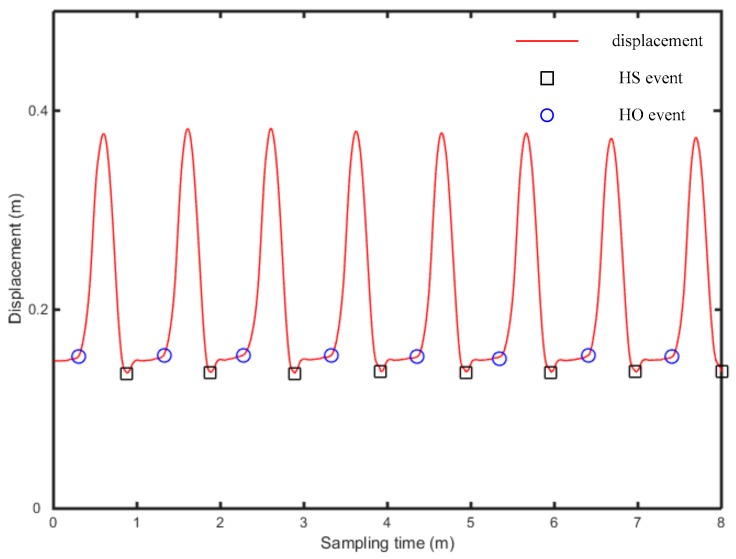
Determination of the heel event.

**Figure 12 sensors-19-05499-f012:**
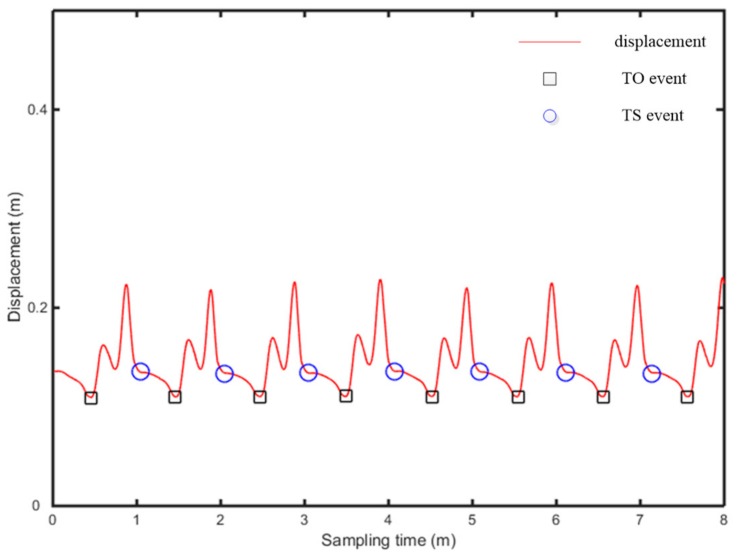
Determination of the toe event.

**Figure 13 sensors-19-05499-f013:**
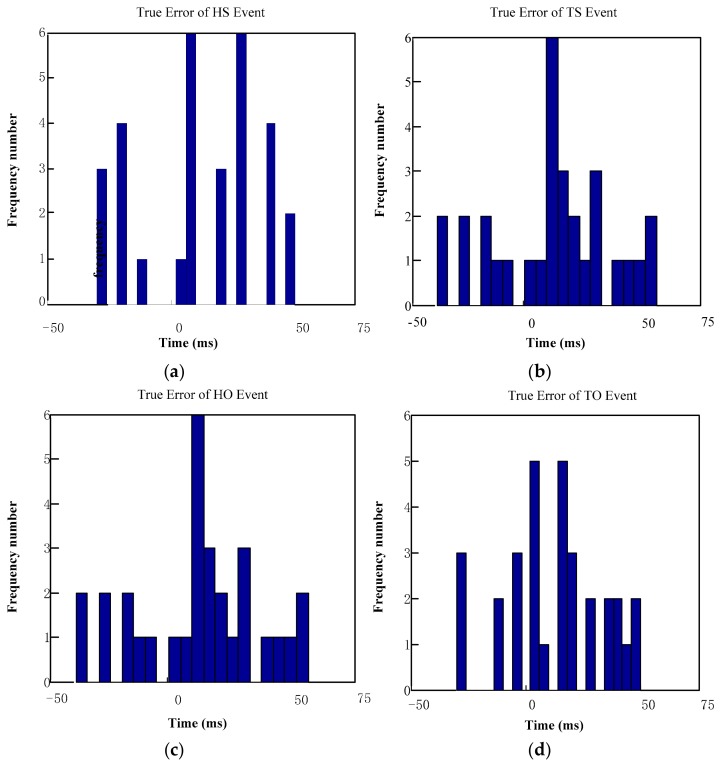
The statistical analysis of error. (**a**) The true error of heel strike (HS) event; (**b**) the true error of toe strike (TS) event; (**c**) the true error of heel off (HO) event; (**d**) the true error of toe off (TO) event.

**Figure 14 sensors-19-05499-f014:**
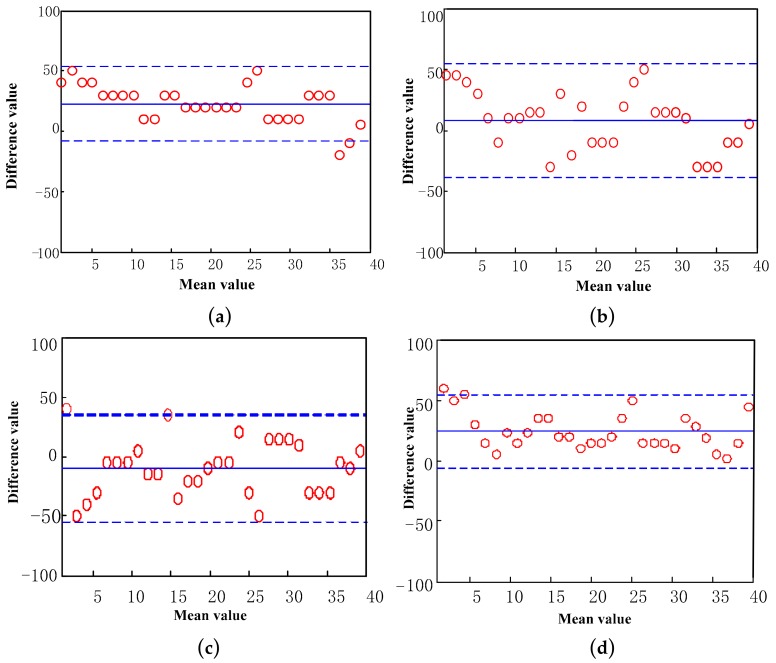
The consistency test of the Bland–Altman graph. (**a**) The Bland–Altman of HS event; (**b**) the Bland–Altman of TO event; (**c**) the Bland–Altman of HO event; (**d**) the Bland–Altman of TO event. The horizontal axis is the mean value of the results of theoretical analysis and the contrast results, and the vertical axis is the difference value of the results of theoretical analysis and the contrast results.

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
