# Peer review of "A Determination Method for Gait Event Based on Acceleration Sensors"

_sensors, 2019, doi:10.3390/s19245499_

Round 1
Reviewer 1 Report
Paper Sensors 652759
The paper theme is interesting and may help the quality of life in several people.
Major review
line 80: The study was not approved by research ethics committee? line 105: Section 2.2, why wavelet, a “simple” Butterworth bandpass filter is not enough?
Minor review
lines 51-56: This phrase is to long, that jeopardize the understandment. line 168: Figure 4, please change the legend in figure to english. line 388: Conclusion, in you introduction you punctuate the viability of system to improve the quality of life, mainly in gait disorders subjects. In this way, how your results can be “easily” incorporate in smartphone of user?
Author Response
The paper theme is interesting and may help the quality of life in several people.
A: Thanks very much for the review’s comments.
Point 1:
Q1: line 80: The study was not approved by research ethics committee? line 105: Section 2.2, why wavelet, a “simple” Butterworth bandpass filter is not enough?
A1: We have submitted the approval file to the Assistant Editor.
A2: Butterworth bandpass filter (BBF) and wavelet modulus maxima algorithm (WMMA) can be used to remove the noise of the raw acceleration signal. In our other study of joint angle estimation, we used BBF to remove noise effectively, but in this study, we find that WMMA, rather than BBF, can not only effectively remove the noise, but also retain the feature peak of the original signal, which is the key to the gait event analysis.
Point 2:
Q2: lines 51-56: This phrase is too long, that jeopardize the understandment. line 168: Figure 4, please change the legend in figure to English. line 388: Conclusion, in you introduction you punctuate the viability of system to improve the quality of life, mainly in gait disorders subjects. In this way, how your results can be “easily” incorporate in smartphone of user?
A1: In revised manuscript, we have rewritten this sentence, as suggested by Reviewer 1.
A2: we have redrawn the Fig.4 and changed the legend in English.
A3: This is a good idea and a potentially important application. In the student competition experiments, we have tried and successfully connected the signal to the mobile phone via the developed APP software. In the future, we also will intend to carry out this essential work.
(In revised manuscript, the color red, green, and blue, respectively, have been used to reply to the comments of Reviewer 1, Reviewer 2, and Reviewer 3.)
Reviewer 2 Report
The authors propose a system to evaluate gait using acceleration sensors. The article is of interest, but I do have some concerns. These are listed hereafter:
Line 11: “widespread use of a clinic” – word formatting issue. The authors should consider the wider literature briefly by considering the advantages and disadvantages of other approaches such as [1]–[3].This helps set the sense as to why wearable sensors are the most appropriate. The authors need to explain why placing the sensors on the toe and heel is appropriate considering the impact on the participant and the practicality/feasibility of undertaking this tracking long term. Why were only 5 participants used in the study? There is a disconnect between the number of subjects used and the claims in the discussion of success. The work should be framed as a feasibility trial based on the sample: no general interface can be made. How generalisation is the gait event determination? It appears that unless the data is recorded in ‘near perfect condition’ the noise may result in misclassification. Threshold value: Did the authors experiment with different threshold values, if so, how did this impact sensitivity and specificity. It would be helpful to present the results in the context of author works and literature. The authors should also describe the pros and cons of wearable sensors.[1] D. Leightley, J. S. McPhee, and M. H. Yap, “Automated Analysis and Quantification of Human Mobility Using a Depth Sensor,” IEEE J. Biomed. Heal. Informatics, vol. 21, no. 4, pp. 939–948, Jul. 2017.
[2] J. Darby, B. Li, and N. Costen, “Tracking a Walking Person using Activity-Guided Annealed Particle Filtering,” in 8th IEEE International Conference on Automatic Face and Gesture Recognition (FG 2008), 2008, pp. 1–6.
[3] B. Galnaa, G. Barrya, D. Jacksonb, D. Mhiripiria, P. Olivierb, and L. Rochestera, “Accuracy of the Microsoft Kinect sensor for measuring movement in people with Parkinson’s disease,” Gait Posture, vol. 39, no. 4, pp. 1062–1068, Apr. 2014.
Author Response
The authors propose a system to evaluate gait using acceleration sensors. The article is of interest, but I do have some concerns. These are listed hereafter:
A: The authors would like to express their greatest appreciation to the Reviewers for their careful review of the paper and, in particular, for their useful and constructive comments and suggestions for the revised version of the paper..
Point 1:
Line 11: “widespread use of a clinic” – word formatting issue. The authors should consider the wider literature briefly by considering the advantages and disadvantages of other approaches such as [1]–[3].This helps set the sense as to why wearable sensors are the most appropriate. The authors need to explain why placing the sensors on the toe and heel is appropriate considering the impact on the participant and the practicality/feasibility of undertaking this tracking long term. Why were only 5 participants used in the study? There is a disconnect between the number of subjects used and the claims in the discussion of success. The work should be framed as a feasibility trial based on the sample: no general interface can be made. How generalisation is the gait event determination? It appears that unless the data is recorded in ‘near perfect condition’ the noise may result in misclassification. Threshold value: Did the authors experiment with different threshold values, if so, how did this impact sensitivity and specificity. It would be helpful to present the results in the context of author works and literature. The authors should also describe the pros and cons of wearable sensors.
A1: we have rewritten this sentence to correct the word formatting issue.
A2: According to the suggestions of Reviewer 2 and Reviewer 3, we revised and expanded the Introduction, contenting of various methods of the pros and cons, technical limitations, etc. [1-3], in order to introduce the study more comprehensively. (Highlighted in green and blue)
A3: From the technical point of view, it is necessary to obtain effective acceleration signals and determine gait events. so need to select the position with the strongest signal for acquisition and analysis. Therefore, this paper used the method of Literature [4] and placed sensors on the toe and heel. Considering the impact on the participant and the practicality/feasibility of undertaking this tracking long term, this is a very useful and practical problem, we haven't done much research at present.
A4: The authors agree that the dataset used in this study is limited as only 5 participants. However, as a proof of concept, this is a preliminary study that aims to propose a method of the gait event determination, i.e., to determine four gait events in the segmented signal during a gait cycle, based on the characteristics of the vertical displacement of heel and toe. In fact, the data of all participants with their gait cycles, were calculated and tested in this method. In addition, the various noises may affect the accuracy of recognition. However, we preprocess the original acceleration signal by denoising process, while detrend the displacement signal obtained by the quadratic integration of acceleration, which reduce the effect of noise so that the errors are acceptable. According to the error analysis and consistency testing, the proposed novel method provides results consistent with those recorded by 3D motion capture system. In this sense, this is an important work for gait recognition using wearable inertial sensors. This framework has been tested for this purpose on gait events, and will be further tested on additional subjects and gait conditions in the future.
A5: The proposed method has two steps: 1. preliminary gait segmentation, 2. gait event determination. Considering the effect of gait events by the threshold value[5], in step 1, we use the threshold of acceleration rate of change, which is less affected by threshold value [6]. In step 2, we propose the algorithm in Section 3.3 to determine the gait event, which does not involve the threshold problem.
Refreences
Galna, B.; Barry, G.; Jackson, D.; Mhiripiri, D.; Olivier, P.; Rochester, L., Accuracy of the Microsoft Kinect sensor for measuring movement in people with Parkinson's disease. Gait & Posture 2014, 39, 1062-1068. Darby, J.; Li, B.; Costen, N. In Tracking a walking person using activity-guided annealed particle filtering, 8th IEEE International Conference on Automatic Face & Gesture Recognition, 17-19 Sept. 2008, 2008;pp 1-6. Leightley, D.; McPhee, J. S.; Yap, M. H., Automated analysis and quantification of human mobility using a depth sensor. IEEE Journal of Biomedical and Health Informatics 2017, 21, 939-948. Mickelborough, J.; van der Linden, M. L.; Richards, J.; Ennos, A. R., Validity and reliability of a kinematic protocol for determining foot contact events. Gait & Posture 2000, 11, 32-37. González I, Fontecha J, Hervás R, et al. Estimation of Temporal Gait Events from a Single Accelerometer Through the Scale-Space Filtering Idea[J]. Journal of Medical Systems, 2016, 40: 251-261. Boutaayamou M, Schwartz C, Stamatakis J, et al. Development and validation of an accelerometer-based method for quantifying gait events[J]. Medical Engineering & Physics, 2015, 37: 226-232.
(In revised manuscript, the color red, green, and blue, respectively, have been used to reply to the comments of Reviewer 1, Reviewer 2, and Reviewer 3.)
Reviewer 3 Report
Review Report – Sensors - 652759
A brief summary
Alternative approach to segmentation and representation of gait cycle was proposed, predominantly based on acceleration signals.
Broad comments
Introducing overview was supported by methodology used to fulfil objectives of the experimental setup. Experimental design was appropriate with adequately presented results and error analysis, followed and supported by relevant and precise conclusions. Additional info about limitations of the study and better presentation of equations are necessary for final version.
Specific comments Equations and formulas should be re-checked – there should be description for each letter even if it is obvious (e.g. in eq.1.- ‘p’), in some descriptions there are wrong letters (in lines 221, 222 letter ‘H’ describes both heel and toe), some equations/formulas do not stand in presented form (e.g. in eq.4 it is not mathematically correct to express it in this form – someone could read that gravitational contribution equals to zero), etc. Please go through all equations! Please add one more sentence about technical limitations of the study (related to footwear, materials, etc.). Express differences in precision, validity, and stability between golden standard (mentioned in line 38) and proposed method (not just lower costs, but more technical details), where possible in percentages.
Author Response
Introducing overview was supported by methodology used to fulfill objectives of the experimental setup. Experimental design was appropriate with adequately presented results and error analysis, followed and supported by relevant and precise conclusions. Additional info about limitations of the study and better presentation of equations are necessary for final version.
A: The authors would like to thank the reviewer for the remarks and advices that helped to improve the original manuscript.
Point 1:
Specific comments Equations and formulas should be re-checked – there should be description for each letter even if it is obvious (e.g. in eq.1.- ‘p’), in some descriptions there are wrong letters (in lines 221, 222 letter ‘H’ describes both heel and toe), some equations/formulas do not stand in presented form (e.g. in eq.4 it is not mathematically correct to express it in this form – someone could read that gravitational contribution equals to zero), etc.
A: Text, charts, equations, etc. were re-checked carefully, while several errors and typos were corrected in revised manuscript.
Point 2:
Please add one more sentence about technical limitations of the study (related to footwear, materials, etc.).
A: According to the suggestions of Reviewer 2 and Reviewer 3, we revised and expanded the Section 1 (Introduction), contenting of various methods of the pros and cons, technical limitations, etc., in order to introduce the study more comprehensively. (Highlighted in green and blue)
Point 3:
Express differences in precision, validity, and stability between golden standard (mentioned in line 38) and proposed method (not just lower costs, but more technical details), where possible in percentages.
A: In revision, we have discussed and compared the differences in technical details between golden standard and the proposed method, which were addressed in Section 1 (Introduction) and Section 5 (Conclusions).
(In revised manuscript, the color red, green, and blue, respectively, have been used to reply to the comments of Reviewer 1, Reviewer 2, and Reviewer 3.)
Round 2
Reviewer 1 Report
I checked the files attached, to me is correct, to paper can be accepted.
Reviewer 2 Report
The author shave addressed my concerns.